# Towards Realistic Earth-Observation Constellation Scheduling: Benchmark and Methodology

**Luting Wang*    Yinghao Xiang*    Hongliang Huang    Dongjun Li    Chen Gao†    Si Liu†**

Beihang University

## Abstract

Agile Earth Observation Satellites (AEOSs) constellations offer unprecedented flexibility for monitoring the Earth's surface, but their scheduling remains challenging under large-scale scenarios, dynamic environments, and stringent constraints. Existing methods often simplify these complexities, limiting their real-world performance. We address this gap with a unified framework integrating a standardized benchmark suite and a novel scheduling model. Our benchmark suite, AEOS-Bench, contains $3,907$ finely tuned satellite assets and $16,410$ scenarios. Each scenario features 1 to 50 satellites and 50 to 300 imaging tasks. These scenarios are generated via a high-fidelity simulation platform, ensuring realistic satellite behavior such as orbital dynamics and resource constraints. Ground truth scheduling annotations are provided for each scenario. To our knowledge, AEOS-Bench is the first large-scale benchmark suite tailored for realistic constellation scheduling. Building upon this benchmark, we introduce AEOS-Former, a Transformer-based scheduling model that incorporates a constraint-aware attention mechanism. A dedicated internal constraint module explicitly models the physical and operational limits of each satellite. Through simulation-based iterative learning, AEOS-Former adapts to diverse scenarios, offering a robust solution for AEOS constellation scheduling. Experimental results demonstrate that AEOS-Former outperforms baseline models in task completion and energy efficiency, with ablation studies highlighting the contribution of each component. Code and data are provided in `https://github.com/buaa-colalab/AEOSBench`.

## 1 Introduction

Agile Earth Observation Satellites (AEOSs) [32, 7, 21] have emerged as a transformative technology in remote sensing, enabling rapid and flexible monitoring of the Earth's surface. By operating co-operatively in constellations [44, 14, 40, 39, 42], multiple AEOSs can dramatically increase revisit frequency and broaden coverage beyond the capability of a single satellite. As shown in Fig. 1, the AEOS constellation scheduling problem seeks to optimally assign imaging tasks across satellites to maximize task completion while minimizing time and resource expenditure [15, 41], all within real-world constraints. Robust scheduling models empower faster and more informed decision-making for applications such as disaster response [5, 26], environmental monitoring [3], and resource management [33].

The challenge of AEOS constellation scheduling stems from three core factors. First, modern constellations may comprise dozens of satellites tasked with hundreds of imaging requests [1]. This

---

*Equal contribution. Email: wangluting@buaa.edu.cn, xiangyinghao@buaa.edu.cn.

†Corresponding authors. Email: gaochen.ai@gmail.com, liusi@buaa.edu.cn.

39th Conference on Neural Information Processing Systems (NeurIPS 2025).

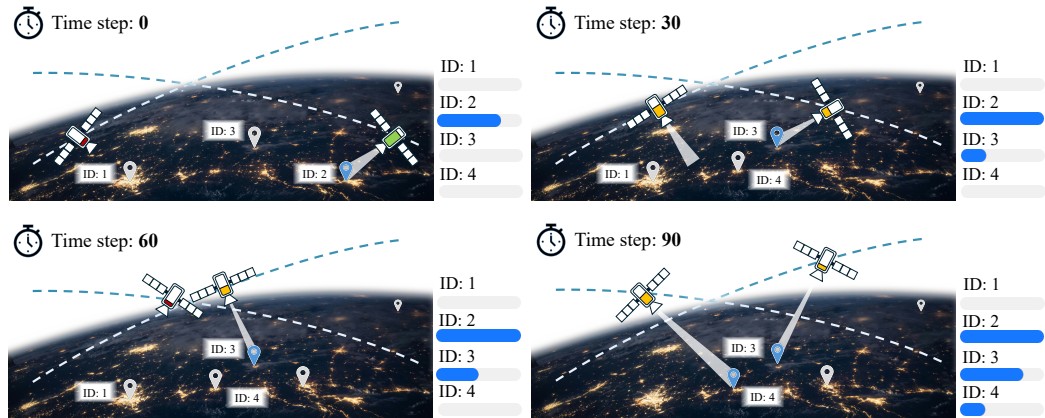

Figure 1: Illustration of AEOS constellation scheduling over four timesteps. At each timestep, satellites adjust their attitude to image ground targets, consuming battery energy while charging via solar panels. Tasks can be published or expired. Multiple satellites can cooperate to complete tasks.

scale renders exhaustive searches infeasible and strains both heuristic algorithms and reinforcement-learning methods [15]. Second, the operating environment is highly dynamic: new tasks can appear or expire at any moment, satellite positions and attitudes are continuously changing, batteries cycle through charge and discharge, and satellites may even join or leave the constellation. Scheduling algorithms must adapt on the fly without foreknowledge of these changes. Third, every assignment of tasks must respect strict constraints, such as the available battery energy, the sensor field of view (FOV), and the allowable time window for each task, or the imaging request cannot be fulfilled.

Any practical scheduling model must simultaneously scale to large constellations, adapt in real time, and respect every operational constraint. However, most existing methods compromise one or more of these goals. For example, REDA [15] is tailored to a fixed set of satellites and tasks under abstracted constraints, while EOSSP-RCS [23] targets small constellations. While effective on simplified benchmarks, their performance degrades sharply in realistic scenarios. Moreover, the absence of a common benchmark prevents fair comparison across scheduling models.

To bridge this gap, we present a unified framework for the AEOS constellation scheduling, comprising a standardized benchmark suite and a novel scheduling model. Our benchmark is built on a simulation platform powered by the Basilisk engine [19], which accurately models each satellite's orbital dynamics, attitude control, and other physical characteristics. We provide $3,907$ satellite assets, each with fine-tuned control parameters to ensure stability during task execution. AEOS-Bench, our benchmark suite, is distinguished by four key features: **1) Large-Scale.** AEOS-Bench includes $16,410$ scenarios, each featuring 1 to 50 satellites, 50 to 300 imaging tasks, and $3,600$ timesteps. **2) Realism.** All scenarios are generated and evaluated on our simulation platform, ensuring physically accurate satellite behavior. The test split incorporates real satellite data from publicly available sources[1], enabling evaluation on authentic data. **3) Comprehensiveness.** AEOS-Bench evaluates six metrics, including task completion rate, turn-around time, and power consumption. **4) Open-Accessible Data.** Every scenario is annotated with ground truth assignments through a rigorous pipeline. All benchmark data and annotations are publicly accessible. To our knowledge, AEOS-Bench is the first large-scale benchmark for realistic AEOS constellation scheduling.

We further introduce AEOS-Former, a Transformer-based [28] scheduler engineered for AEOS constellations. At its core lies a dedicated internal constraint module that explicitly models each satellite's physical and operational limits, including sensor field of view, battery state, and attitude control time. By predicting a feasibility probability and minimal control time, this module produces a constraint-driven attention mask to guide scheduling. AEOS-Former begins by embedding static attributes (*e.g.*, orbital parameters, target location) and dynamic states (*e.g.*, current attitude, task progress). A transformer encoder ingests task embeddings to produce contextual task features. Concurrently, the decoder takes satellite embeddings and attends to the task features under the constraint

---

[1]N2YO (www.n2yo.com) and Gunter's Space Page (space.skyrocket.de).

Table 1: Comparison of existing benchmarks. Our AEOS-Bench incorporates 16k scenarios with realistic physics simulator and ground truth annotations.

| Setting | Benchmark | #Scene | #Sat | #Task | Traj. Len. | Phy. Sim. | Ann. |
|---|---|---|---|---|---|---|---|
| Single Satellite | Eddy *et al.* [8] | 30 | 1 | 200~2000 | 500s | ✗ | ✗ |
| | Herrmann *et al.* [14] | 45k | 1 | 135 | 4.5h | ✓ | ✗ |
| | H-PPO [34] | 5 | 1 | 100~2000 | 30m | ✗ | ✗ |
| | TRM-TE [23] | 100k | 1 | 50~200 | - | ✗ | ✗ |
| Multiple Satellites | EHE-DCF [36] | 8 | 10 | 200~1600 | 1h | ✗ | ✗ |
| | SFMODBO [30] | 4 | - | 50~200 | 3h | ✗ | ✗ |
| | SatNet [11] | 5 | 29~33 | 257~333 | 168h | ✗ | ✗ |
| | REDA [15] | 1 | 324 | 450 | 100m | ✗ | ✗ |
| | AEOS-Bench (Ours) | 16k | 1~50 | 50~300 | 1h | ✓ | ✓ |

mask, yielding an assignment matrix. To extend beyond purely supervised learning, AEOS-Former is integrated in a simulation-based iterative learning loop. After pretraining on AEOS-Bench annotations, it is deployed in our simulator to explore random scenarios. Schedules exceeding a preset performance threshold are merged back into AEOS-Bench for retraining. Through iterative cycles of constraint-driven attention and simulator-guided exploration, AEOS-Former converges on high-value scheduling strategies that generalize across diverse scenarios.

To evaluate the effectiveness of AEOS-Former, we conduct a series of comparison experiments against several baseline models, using six metrics that encompass task completion, timeliness, and energy efficiency. On the val-unseen split, AEOS-Former achieves a completion rate of $35.42\%$, with a power consumption of only 68.99 Wh, outperforming the baseline ($35.35\%$ completion rate and 140.83 Wh power consumption). Moreover, AEOS-Former surpasses all baselines across all splits in terms of the comprehensive score. Ablation studies further confirm the contribution of each component in AEOS-Former. By providing the AEOS-Bench and AEOS-Former, we hope this work will inspire novel methods in AEOS constellation scheduling.

## 2 Related Work

To solve the constellation scheduling problem, researchers have developed various benchmarks and methods. Methods can be broadly classified as optimization-based or neural-network-based.

**Benchmarks.** As summarized in Tab. 1, most existing benchmarks for multi-satellite scheduling include fewer than 10 scenarios, limiting their diversity and generalizability. In contrast, AEOS-Bench offers $16,410$ diverse scenarios. Unlike prior benchmarks, AEOS-Bench further leverages a high-fidelity simulation platform with expert-generated ground truth annotations. These features ensure both realistic constrains and reliable evaluation metrics for real-world applicability.

**Optimization-based Methods.** Early studies rely on exact solvers to optimize satellite assignments. Lemaître *et al.* [21] adopt a constraint programming framework for agile satellite scheduling. Sin *et al.* [27] uses sequential convex programming to accelerate target acquisition. Although these methods guarantee optimality, their computational cost escalates sharply with the problem scale. Subsequent heuristic methods aim to improve scalability [31, 6, 12, 37, 38, 25]. HAAL [16] balances performance and runtime via handover-aware task allocation. MSCPO-SHCS [9] employs a stochastic hill-climbing strategy for timely assignment optimization. Other approaches include Ant Colony Optimization [17], evolutionary algorithm [10], and genetic algorithm [2]. While these methods offer faster runtimes, their performance diminishes with large-scale or dynamic scenarios.

**Neural-Network-based Methods.** The robust fitting capabilities of neural networks have driven breakthroughs across diverse domains [4, 20, 22], including constellation scheduling [35, 43]. Herrmann *et al.* [13] formulates the problem as a Markov decision process (MDP) and adopts reinforcement learning for scheduling. Pointer Networks [29] provide a sequence-to-sequence formulation for combinatorial assignments. EOSSP-RCS [23] proposed a Transformer-based encoder–decoder architecture with temporal encoding model and achieved relatively good performance. Infantes *et al.* [18] adopts GNN and Deep Reinforcement Learning to the Earth Observation Satellite Planning problem with very competitive performance. REDA [15] combines multi-agent RL with

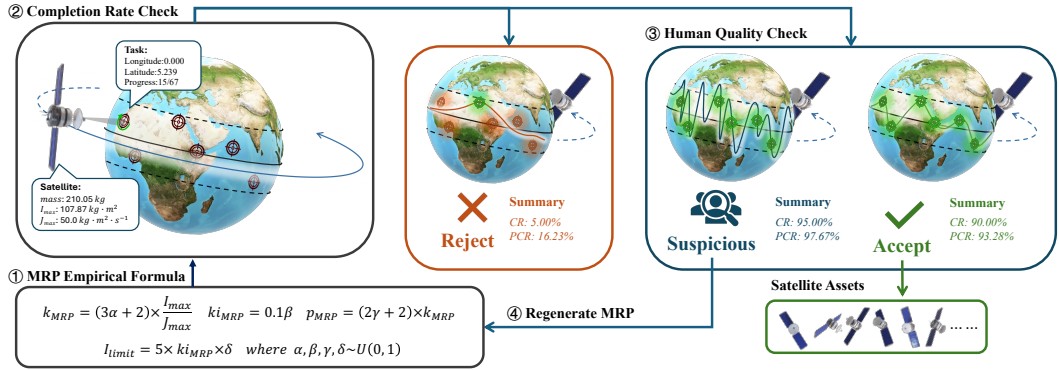

Figure 2: The generation process of satellite assets, which incorporates an empirical formula and multiple checks to ensure stable attitude control for each asset.

polynomial-time greedy solvers to balance assignment quality and speed. Despite promising results, many of these methods simplify key physical constraints. In contrast, our AEOS-Former integrates an intrinsic constraint module that explicitly enforces physical and operational limitations, substantially improving the feasibility and fidelity of generated schedules.

# 3 The AEOS-Bench Suite

In this section, we first define the problem setup of AEOS constellation scheduling. Next, we describe the process of generating satellite assets and ground truth scheduling annotations for AEOS-Bench. Finally, we provide an analysis of AEOS-Bench.

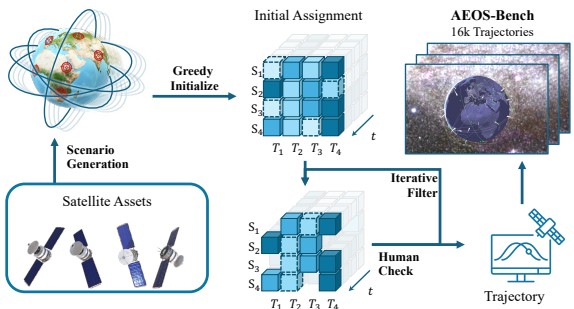

Figure 3: The annotation pipeline for AEOS-Bench.

## 3.1 Problem Setup

**Scenario Modeling.** To capture the essential physics that determines task feasibility, we model each satellite as a composition of four core subsystems: orbital dynamics, attitude control, power system, and sensor payload. Satellites occupy low-Earth orbit (LEO), with parameters like orbital elements, mass properties, and moments of inertia sampled uniformly from representative ranges (details in Appendix C). Attitude control employs the Modified Rodrigues Parameters (MRP) formalism, with control gains and acutator limits specified per satellite in Sec. 3.2. We collect the satellite characteristics into a matrix $\mathbf{S}^s \in \mathbb{R}^{N_S \times d_S^s}$, where $N_S$ denotes the number of satellites and $d_S^s$ the feature dimension. Imaging tasks arrive dynamically, each defined by a release time, due time, required observation duration, and the ground-target coordinates. These task descriptors form a matrix $\mathbf{T}^s \in \mathbb{R}^{N_T \times d_T^s}$, with $N_T$ tasks and $d_T^s$ task attributes.

**Action Space.** We adopt a two-tier action abstraction to separate high-level scheduling from low-level control. The low-level action space comprises power-on/off commands and attitude-pointing directives, which are dispatched directly to the Basilisk engine to simulate battery cycling, sensor activation, and MRP-based attitude maneuvers. While this affords maximal control flexibility, it imposes excessive complexity on scheduling models. Instead, our high-level action space consists of task-assignment commands. The scheduler outputs an assignment vector $a = [a_1, a_2, \ldots, a_{N_S}]$, where each $a_i \in \{0, 1, \ldots, N_T\}$. A value of $a_i = 0$ directs satellite $i$ to power down its sensor, while any $a_i > 0$ instructs it to activate the sensor and reorient to service task $a_i$. The platform automatically converts these high-level assignments into low-level commands, allowing scheduling models to concentrate purely on task selection and timing.

**Constraints.** Real-world AEOS constellation scheduling is governed by multiple constraints. We enforce 5 constraints in our platform: dynamics, energy, FOV, continuity, and time window (de-

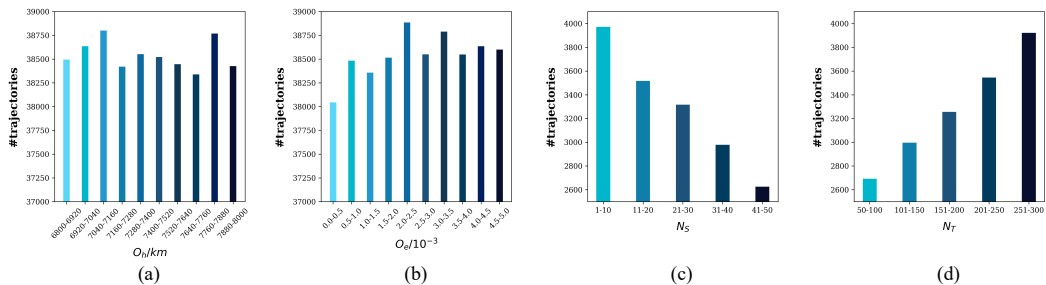

Figure 4: Statistical analysis of AEOS-Bench. (a) and (b) show the distribution of trajectories w.r.t. the semi-major axis and eccentricity of satellite orbits, respectively. (c) and (d) illustrate the distribution of trajectories w.r.t. the number of satellites and tasks, respectively.

tails in Appendix C). Any high-level assignment that violates these constraints is rejected by the simulator, and only successful observations are recorded for downstream benchmarking.

## 3.2 Data Collection

The attitude control system in our simulation platform uses the MRP method, whose performance relies on several key parameters: control gains and actuator limits. These parameters govern the speed and precision that a satellite can adjust its attitude. Low control gains result in slow attitude adjustments, while overloaded actuators can destabilize the satellite, risking task failures. To ensure dependable performance under these conditions, we repeat the cycle in Fig. 2 until we accumulate $3,907$ satellite assets, each proven to deliver reliable on-orbit performance.

While our platform supports closed-loop simulation, training scheduling models from scratch via simulator roll-outs is computationally expensive. To bootstrap learning, we curated AEOS-Bench: a large dataset with constellation scheduling annotations. As shown in Fig. 3, each AEOS-Bench scenario begins with a distance-based initialization. While simple and intuitive, this method often assigns tasks that lie too close to the satellite, leading to attitude control failures. Therefore, we introduce the iterative filter stage and human quality review. Through this process, AEOS-Bench delivers reliable scheduling data for training schedulers.

## 3.3 Data Analysis

We partition AEOS-Bench into four splits. The train split consists of $16,218$ trajectories with $2,907$ satellite assets. The val-seen split includes $64$ scenarios using the same satellites as the train split. The val-unseen split features $64$ scenarios with $500$ satellites not present in the train split. The test split contains $64$ scenarios with $500$ satellites, each having realistic properties sourced from the web.

As shown in Fig. 4, the orbital parameters of each satellite asset follow an approximately random distribution within specific ranges. Scenarios with smaller constellations or a larger number of tasks are more frequent in AEOS-Bench. This may be because generating high-quality assignments is easier when there are fewer satellites and more tasks.

## 4 The AEOS-Former Model

This section begins with the dynamic data processing pipeline in Sec. 4.1. Next, Sec. 4.2 introduces our internal constraint module for the prediction of feasibility and control time. In Sec. 4.3, we detail the transformer-based satellite–task matching architecture. Finally, Sec. 4.4 presents our simulation-driven iterative learning pipeline. The architecture of our AEOS-Former is demonstrated in Fig. 5.

## 4.1 Dynamic Data Processing

As demonstrated in Sec. 3.1, each scenario in the AEOS-Bench is defined by a static satellite matrix $\mathbf{S}^s$ and a static task matrix $\mathbf{T}^s$, which capture time-independent properties. Dynamic properties, such as task progress and satellite attitude, are not contained within these static matrices. Enabling

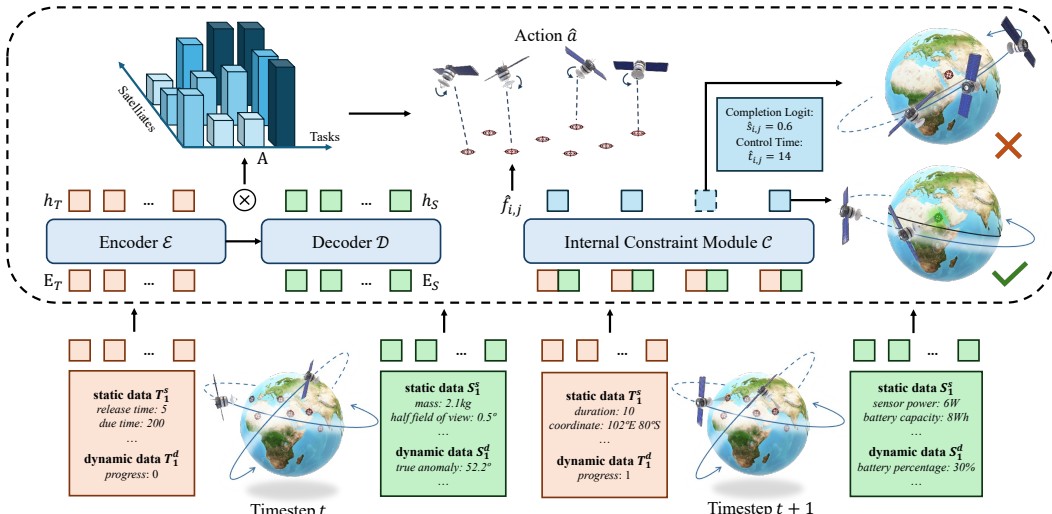

Figure 5: Architecture of AEOS-Former. Static and dynamic data of satellites and tasks are first concatenated and embedded. A transformer encoder processes task features, and a decoder attends to satellite embeddings under a constraint-derived cross-attention mask. The internal constraint module then predicts feasibility logits and required control times, guiding action selection.

the scheduling model to infer dynamic states from static properties and past decisions would substantially increase complexity without clear benefit. Instead, we query our simulator at each timestep to retrieve the current dynamic satellite and task properties.

The full input matrices are formed by concatenating static and dynamic components:

$$\mathbf{S} = \left[\mathbf{S}^s; \mathbf{S}^d\right] \in \mathbb{R}^{N_S \times d_S}, \qquad \mathbf{T} = \left[\mathbf{T}^s; \mathbf{T}^d\right] \in \mathbb{R}^{N_T \times d_T}, \tag{1}$$

where $\mathbf{S}^d \in \mathbb{R}^{N_S \times d_S^d}$ is the dynamic satellite matrix, $\mathbf{T}^d \in \mathbb{R}^{N_T \times d_T^d}$ is the dynamic task matrix, $d_S = d_S^s + d_S^d$, and $d_T = d_T^s + d_T^d$.

To embed temporal context into AEOS-Former, we incorporate a sinusoidal time embedding $\mathbf{E}_t$ at the current timestep $t$. Task release and due times are converted into relative time offsets w.r.t. $t$. Finally, we normalize both $\mathbf{S}$ and $\mathbf{T}$ using statistics computed over the entire AEOS-Bench dataset.

### 4.2 The Internal Constraint Module

To explicitly model the constraints inherent in our platform, we introduce an internal constraint module $\mathcal{C}$. For a satellite-task pair $(i, j)$, $\mathcal{C}$ predicts the feasibility of satellite $i$ performing task $j$:

$$\hat{f}_{i,j} = \mathcal{C}([\mathbf{S}_i; \mathbf{T}_j]), \quad 1 \leq i \leq N_S, \quad 1 \leq j \leq N_T, \tag{2}$$

where $\hat{f}_{i,j} = \begin{bmatrix} \hat{s}_{i,j} & \hat{t}_{i,j} \end{bmatrix} \in \mathbb{R}^2$ comprises two components: $\hat{s}_{i,j}$ is the predicted logit indicating the feasibility of satellite $i$ completing task $j$, and $\hat{t}_{i,j}$ is the estimated time for attitude adjustment.

Ideally, ground truth labels $s_{i,j} \in \{0, 1\}$ would be available to supervise $\hat{s}_{i,j}$. However, in AEOS-Bench, many tasks are accomplished through the collaboration of multiple satellites, making it challenging to attribute task completion to individual satellites directly. Determining $s_{i,j}$ would necessitate dedicated simulations, which are computationally intensive.

To address this, we define an approximate label $\tilde{s}_{i,j} \in \{0, 1\}$, which can be easily obtained from AEOS-Bench. We set $\tilde{s}_{i,j} = 1$ if satellite $i$ contributed to task $j$ for at least $n$ consecutive timesteps and the task is completed in the trajectory. The loss function is defined using binary cross-entropy:

$$\mathcal{L}_s = \frac{1}{N_S N_T} \sum_{i=1}^{N_S} \sum_{j=1}^{N_T} \text{BCE}(\hat{s}_{i,j}, \tilde{s}_{i,j}). \tag{3}$$

To further guide $\mathcal{C}$ in internalizing constraints, we introduce time supervision. If $\tilde{s}_{i,j} = 1$, we denote $\tilde{t}_{i,j}$ as the minimal time offset $\Delta t$ from the current timestep $t$ such that satellite $i$ begins continuous

contribution to task $j$. The corresponding loss function is:

$$\mathcal{L}_t = \sum_{i=1}^{N_S} \sum_{j=1}^{N_T} \tilde{s}_{i,j} \cdot \text{MSE}(\hat{t}_{i,j}, \tilde{t}_{i,j}) \Big/ \sum_{i=1}^{N_S} \sum_{j=1}^{N_T} \tilde{s}_{i,j}. \tag{4}$$

This dual supervision strategy enables $\mathcal{C}$ to learn both the feasibility and temporal aspects of satellite-task assignments, effectively capturing the constraints present in AEOS-Bench scenarios.

## 4.3 Satellite-Task Matching

To match satellites with tasks, we employ an encoder-decoder architecture that jointly processes satellite and task embeddings, guided by our internal constraint module.

First, we project $\mathbf{S}$ and $\mathbf{T}$ into embedding space and append a sinusoidal timestep embedding $\mathbf{E}_t$:

$$\mathbf{E}_S = [\mathcal{E}_S(\mathbf{S}); \mathbf{E}_t], \qquad \mathbf{E}_T = [\mathcal{E}_T(\mathbf{T}); \mathbf{E}_t], \tag{5}$$

where $\mathcal{E}_S$ and $\mathcal{E}_T$ are the embedding modules. Categorical data (*e.g.*, sensor modes) are looked up in embedding matrices, while continuous ones (*e.g.*, mass, progress) use linear projections.

We encode task features with a transformer encoder $\mathcal{E}$: $h_T = \mathcal{E}(\mathbf{E}_T)$. Then, we decode satellite features via a transformer decoder $\mathcal{D}$, attending to tasks under a mask $\mathbf{M}$: $h_S = \mathcal{D}(\mathbf{E}_S, h_T, \mathbf{M})$. The cross-attention mask $\mathbf{M} \in \mathbb{R}^{N_S \times N_T}$ is derived from the constraint logits: $\mathbf{M}_{i,j} = w \times \hat{s}_{i,j} + b$, with $w, b$ initialized to 0 for stable training. We compute an assignment score matrix $\mathbf{A}$:

$$\mathbf{A} = h_S \cdot [h_\phi; h_T]^\top \in \mathbb{R}^{N_S \times (1+N_T)}, \tag{6}$$

where $h_\phi$ is a trainable vector representing the null assignment. The loss function is defined as:

$$\mathcal{L}_a = \frac{1}{N_S N_T} \sum_{i=1}^{N_S} \sum_{j=1}^{N_T} \text{CE}(\mathbf{A}, a+1), \tag{7}$$

where $a$ is the ground truth assignments in Sec. 3.1. At test time, we filter out infeasible pairs via the constraint logits before sampling from $\mathbf{A}$:

$$\hat{a}_i = -1 + \arg\max_{1 \le j \le N_T} \mathbb{1}\{\sigma(\hat{s}_{i,j}) > \tau_s\} \cdot \mathbf{A}_{i,j}, \tag{8}$$

where $\mathbb{1}\{\cdot\}$ is the indicator function, $\sigma$ is the sigmoid function, $\hat{s}_{i,j}$ is the predicted logits of task completion, and $\tau_s$ is a predefined feasibility threshold. This design tightly integrates learned constraints with feature matching, enabling efficient satellite–task assignments.

## 4.4 Simulation-based Iterative Learning

To fully leverage our simulator platform, we introduce an iterative learning pipeline as demonstrated in Fig. 6.

In the supervised pretraining stage, we initialize AEOS-Former with random weights and train it on the annotated trajectories in AEOS-Bench. The overall loss is a weighted sum of feasibility, timing, and assignment objectives:

$$\mathcal{L} = w_s \cdot \mathcal{L}_s + w_t \cdot \mathcal{L}_t + w_a \cdot \mathcal{L}_a, \tag{9}$$

where $w_s$, $w_t$, and $w_a$ balance the three components. This stage bootstraps the model with basic scheduling strategies learned from expert annotations.

In the subsequent simulation-driven exploration stage, we generate new scenarios and use the pretrained AEOS-Former to propose schedules. Each generated trajectory is evaluated by a comprehensive score as defined in Eq. (10). We then collect only those trajectories whose performance exceeds a predefined threshold $\tau_e$. These high-quality schedules are added back into the AEOS-Bench training set. We repeat this loop until convergence. In this way, AEOS-Former continually refines its policy, discovering novel strategies beyond the original annotations and adapting to increasingly diverse scenarios.

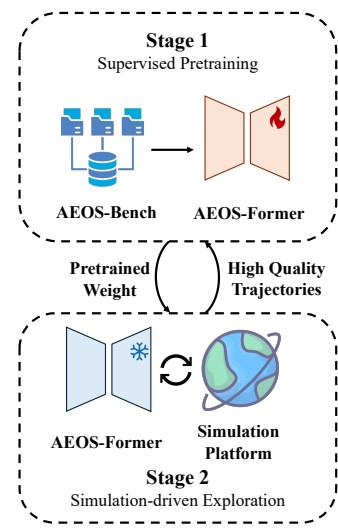

Figure 6: The iterative learning framework with two stages: supervised pretraining and simulation-driven exploration.

Table 2: Performance Comparison between AEOS-Former and Baseline Scheduling Models.

| Split | Method | CS $\downarrow$ | CR/% | PCR/% | WCR/% | TAT/h $\downarrow$ | PC/Wh $\downarrow$ |
|---|---|---|---|---|---|---|---|
| Val Seen | Random | 116.81 | 0.83 | 0.99 | 0.83 | **0.20** | 136.92 |
| | HAAL [16] | 101.09 | 0.98 | 1.09 | 0.97 | 0.23 | 148.02 |
| | REDA [15] | 31.60 | 3.22 | 3.80 | 3.15 | 0.74 | 147.09 |
| | MSCPO-SHCS [9] | 5.85 | 28.77 | 32.93 | 28.23 | 7.75 | 135.93 |
| | AEOS-Former (Ours) | **5.00** | **30.47** | **33.68** | **30.05** | 7.50 | **71.27** |
| Val Unseen | Random | 90.27 | 1.08 | 1.33 | 1.02 | **0.17** | 142.27 |
| | HAAL [16] | 77.17 | 1.28 | 1.46 | 1.28 | 0.25 | 155.36 |
| | REDA [15] | 21.54 | 4.83 | 5.75 | 4.85 | 0.71 | 153.95 |
| | MSCPO-SHCS [9] | 5.21 | 35.35 | **39.45** | 34.85 | 7.27 | 140.83 |
| | AEOS-Former (Ours) | **4.43** | **35.42** | 38.93 | **35.14** | 6.78 | **68.99** |
| Test | Random | 113.53 | 0.85 | 1.02 | 0.88 | **0.17** | 150.54 |
| | HAAL [16] | 94.83 | 1.05 | 1.17 | 1.03 | 0.25 | 155.56 |
| | REDA [15] | 28.21 | 3.65 | 4.27 | 3.58 | 0.73 | 154.49 |
| | MSCPO-SHCS [9] | 7.33 | **19.44** | **24.00** | 18.71 | 6.23 | 149.20 |
| | AEOS-Former (Ours) | **6.28** | 19.25 | 22.31 | **18.73** | 5.67 | **40.91** |

## 5 Experiments

This section begins with the implementation details of AEOS-Former. Next, we introduce the metrics used to evaluate AEOS-Former and baselines. Sec. 5.3 presents the comparison experiments and ablation studies. Sec. 5.4 provides a performance analysis of AEOS-Former through visualization.

### 5.1 Implementation Details

The internal constraint module $\mathcal{C}$ is implemented as a multi-layer perception (MLP) with two hidden layers of width $1024$. The transformer encoder $\mathcal{E}$ and decoder $\mathcal{D}$ are configured with a width of $512$, a depth of $12$, and $16$ attention heads. All loss weights are assigned as $w_s = w_t = w_a = 1$.

Training is conducted with the AdamW optimizer [24] with a base learning rate of $10^{-4}$, $\beta_1 = 0.9$, $\beta_2 = 0.98$, and weight decay $10^{-4}$. Each training batch contains $48$ timesteps uniformly sampled from a trajectory. The supervised stage spans $30,000$ iterations, with a linear warm-up of the learning rate from $10^{-8}$ to $10^{-4}$ over the first $10,000$ iterations. The complete iterative pipeline comprises three supervised stages, culminating in a total of $90,000$ iterations.

Both training and evaluation are performed on a Linux server with 256 CPU cores, 984 GB RAM, and 8 RTX 4090 GPUs. The training process demands approximately 48 GPU-hours. Evaluation is executed over 96 parallel simulator environments and completes in about 30 minutes.

### 5.2 Evaluation Metrics

We evaluate schedulers using six metrics including task completion, timeliness, and energy efficiency. Completion rate (CR) measures the proportion of completed tasks out of all. Partial completion rate (PCR) assesses the ratio of the maximum progress to the total required duration. Weighted completion rate (WCR) is a weighted version of CR, considering task durations. Turn-around time (TAT) calculates the average time taken to complete tasks, reflecting scheduling efficiency. Power consumption (PC) quantifies the total energy consumed by the satellite sensors during imaging. Finally, the comprehensive score (CS) aggregates these metrics into a single performance indicator:

$$\text{CS} = (w_{\text{CR}} \cdot \text{CR} + w_{\text{PCR}} \cdot \text{PCR} + w_{\text{WCR}} \cdot \text{WCR})^{-1} + w_{\text{TAT}} \cdot \text{TAT} + w_{\text{PC}} \cdot \text{PC}, \qquad (10)$$

where $w_{\text{CR}} = 0.6$, $w_{\text{PCR}} = 0.2$, $w_{\text{WCR}} = 0.2$, $w_{\text{TAT}} = 1/7$, and $w_{\text{PC}} = 1/100$.

### 5.3 Main Results

We benchmark our AEOS-Former with several scheduling models. HAAL and MSCPO-SHCS are optimization-based scheduling models, while REDA adopts the multi-agent reinforcement learn-

Table 3: Ablation study on AEOS-Former.

| Split | Constraint Module $\mathcal{C}$ | Iterative Training | CS↓ | CR/% | PCR/% | WCR/% | TAT/h↓ | PC/Wh↓ |
|---|---|---|---|---|---|---|---|---|
| Val Seen | | | 5.85 | 27.47 | 30.88 | 27.16 | **6.56** | 135.94 |
| | ✓ | | 5.27 | 28.06 | 30.84 | 27.82 | 7.54 | **69.76** |
| | | ✓ | 5.28 | **34.25** | **38.04** | **33.80** | 7.44 | 135.90 |
| | ✓ | ✓ | **5.00** | 30.47 | 33.68 | 30.05 | 7.50 | 71.27 |
| Val Unseen | | | 5.17 | 34.05 | 37.83 | 33.60 | **6.21** | 140.84 |
| | ✓ | | 4.51 | 33.71 | 36.79 | 33.57 | 6.43 | **67.84** |
| | | ✓ | 4.72 | **40.88** | **46.72** | **40.58** | 6.55 | 140.87 |
| | ✓ | ✓ | **4.43** | 35.42 | 38.93 | 35.14 | 6.78 | 68.99 |
| Test | | | 9.31 | 13.26 | 15.83 | 12.92 | **3.67** | 149.28 |
| | ✓ | | 7.02 | 16.44 | 18.64 | 16.30 | 5.11 | **36.57** |
| | | ✓ | 6.29 | **24.67** | **28.85** | **24.21** | 6.01 | 149.26 |
| | ✓ | ✓ | **6.28** | 19.25 | 22.31 | 18.73 | 5.67 | 40.91 |

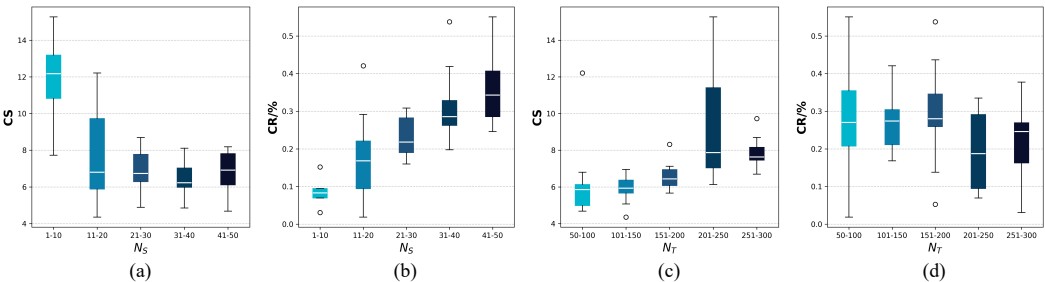

Figure 7: Distribution of CS and CR metrics across varying values of $N_S$ and $N_T$ on the test split.

ing approach. These models were originally designed for simplified environments and do not directly accommodate the comprehensive constraints of our AEOS-Bench setup. Therefore, we have adapted their formulations to ensure compatibility. Additionally, we include a random scheduling model to provide a baseline performance measure. As shown in Tab. 2, AEOS-Former outperforms all baselines across all splits. Notably, on the test split, AEOS-Former achieves $6.28$ CS, surpassing MSCPO-SHCS by $16.7\%$. Thanks to our integrated constraint module and iterative learning paradigm, our model achieves a better balance between CR and PC.

To assess the impact of each component within AEOS-Former, an ablation study is conducted, as shown in Tab. 3. On the val-seen split, incorporating the constraint module enhances both CR and PC, increasing CR from $27.47$ to $28.06$ and reducing PC from $135.94$ to $69.76$. Iterative training further boosts CR to $30.47$. Due to the conflict between CR and PC, the final CR is lower than the CR achieved by sole iterative training. Nonetheless, the CS still improves by more than $0.27$.

## 5.4 Analysis

The baselines include both optimization-based and learning-based methods. Specifically, HAAL and MSCPO-SHCS are optimization-based approaches, while REDA is a neural network-based method.

As shown in Fig. 7, as $N_S$ increases from $1$ to $50$, the CS metric initially decreases before stabilizing between $31$ and $40$, while the CR metric consistently increases. This suggests a

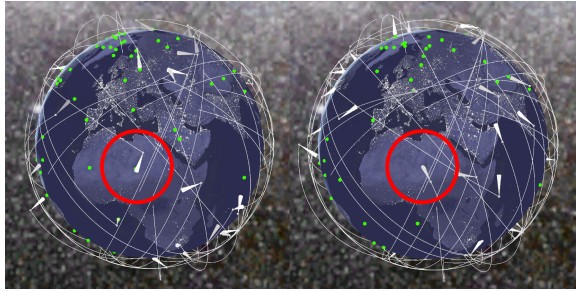

Figure 8: Scheduling visualization of AEOS-Former.

trade-off between task completion and resource consumption. Regarding $N_T$, an increase in the number of tasks leads to lower completion rates and more resource consumption, with the CS metric slightly increasing and the CR metric slightly decreasing.

We also visualize the scheduling of AEOS-Former with Unity3D. In the highlighted areas of Fig. 8, satellite collaborations are observed.

# 6 Conclusion

This work introduces a comprehensive framework for Agile Earth Observation Satellites (AEOS) constellation scheduling. We present AEOS-Bench, a standardized benchmark with $3,907$ satellite assets and $16,410$ scenarios, enforcing realistic constraints and providing ground truth annotations. To our knowledge, AEOS-Bench is the first large-scale benchmark for realistic constellation scheduling. We also propose AEOS-Former, a Transformer-based scheduler featuring a novel constraint module. Through simulation-based iterative learning, AEOS-Former outperforms baselines across diverse scenarios, with ablation studies validating the effectiveness of each component. We hope AEOS-Bench and AEOS-Former will drive innovations in AEOS constellation scheduling.

## Acknowledgement

This research is supported in part by National Key R&D Program of China (2022ZD0115502), National Natural Science Foundation of China (No. 62461160308, U23B2010), "Pioneer" and "Leading Goose" R&D Program of Zhejiang (No. 2024C01161), Beijing Natural Science Foundation (QY25227), Ningbo Science and Technology Innovation 2025 Major Project (2025Z034), NSFC-RGC Project (N_CUHK498/24).

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

Figure 9: Architecture of the simulation platform used in AEOS-Bench.

# A Limitations

In our AEOS-Bench, each task is represented as a single location point. In the future, we plan to propose a new task to incorporate area-based task representations, allowing each observation request to span a defined region. This would enable the evaluation of scheduling algorithms under more realistic constraints, such as partial area coverage, time-window flexibility, and spatial prioritization.

# B Broader Impacts

AEOS-Bench is an open-source suite for AEOS constellation scheduling research, enabling researchers to develop more effective models and conduct fair comparisons. Enhanced scheduling models for AEOS constellations offer several societal benefits. In disaster response, optimized task assignment delivers timely data to first responders, improving search and rescue operations, damage assessment, and resettlement planning. In environmental protection, high-quality imagery data enables early detection of threats such as illegal logging and industrial pollution, strengthening ecosystem oversight and facilitating rapid intervention.

# C Scenario Modeling

Fig. 9 demonstrates the architecture of our simulation platform. The green modules simulate satellite components, including reaction wheels, batteries, sensors, and solar panels. Reaction wheels and sensors draw power from batteries, while solar panels recharge those batteries. The blue modules handle satellite dynamics. A planetary environment, including the Sun and the Earth, supplies the solar incidence angle for the solar panels and simulates gravitational forces. Closed-loop attitude control is achieved by the navigation module, the attitude-guiding module, the MRP control module, and the reaction-wheel control module. The MRP algorithm adjusts the orientation of satellites to keep target locations in view.

Parameters for the simulation platform are summarized in Tab. 4. The range for each parameter is also specified to facilitate random scenario generation. Task parameters are listed in Tab. 5.

Table 4: Satellite parameters.

| Index | Description | Range | Unit |
|-------|-------------|-------|------|
| 1 | scaled moment of inertia | $50 \cdot \mathbf{I}_3 \sim 200 \cdot \mathbf{I}_3$ | $\mathrm{kg} \cdot \mathrm{m}^2$ |
| 2 | scaled mass | $50 \sim 200$ | kg |
| 3 | direction of solar panel | $-180 \sim 180$
$-90 \sim 90$
$-180 \sim 180$ | deg |
| 4 | scaled area of solar panel | $5 \sim 10$ | $\mathrm{m}^2$ |
| 5 | half field of view (FOV) of sensor | $0.5 \sim 1.5$ | rad |
| 6 | power of sensor | $2 \sim 8$ | W |
| 7 | power status of sensor | $\{0, 1\}$ | - |
| 8 | battery capacity | $8{,}000 \sim 30{,}000$ | $\mathrm{mA} \cdot \mathrm{h}$ |
| 9 | battery percentage | $0 \sim 100$ | % |
| 10 | maximum angular momentum of reaction wheels | $10 \sim 100$ | $\mathrm{kg} \cdot \mathrm{m}^2/\mathrm{s}$ |
| 11 | direction of reaction wheels | $-180 \sim 180$
$-90 \sim 90$
$-180 \sim 180$ | deg |
| 12 | angular speed of reaction wheels | $-6{,}000 \sim 6{,}000$ | rpm |
| 13 | power of reaction wheels | $0 \sim 22$ | W |
| 14 | power efficiency of reaction wheels | $0.1 \sim 0.5$ | - |
| 15 | MRP control parameter k
MRP control parameter ki
MRP control parameter p
MRP control parameter integral limit | $2 \sim 5$
$0.0 \sim 0.1$
$6 \sim 12$
$0.0 \sim 0.5$ | - |
| 16 | orbital true anomaly | $0 \sim 360$ | deg |
| 17 | orbital eccentricity | $0 \sim 0.005$ | - |
| 18 | orbital semi-major axis length | $6{,}800 \sim 8{,}000$ | km |
| 19 | orbital inclination | $0 \sim 180$ | deg |
| 20 | orbital right ascension of the ascending node | $0 \sim 360$ | deg |
| 21 | orbital argument of perigee | $0 \sim 360$ | deg |

Table 5: Task parameters

| Index | Description | Range | Unit |
|-------|-------------|-------|------|
| 1 | minimum time of consecutive observation for a task to be considered completed | $15 \sim 60$ | s |
| 2 | release time | $0 \sim 3{,}600$ | s |
| 3 | due time | $0 \sim 3{,}600$ | s |
| 4 | latitude of the target location
longitude of the target location | $-90 \sim 90$
$-180 \sim 180$ | deg
deg |

