# OpenReview forum: "Towards Realistic Earth-Observation Constellation Scheduling: Benchmark and Methodology"
_NeurIPS.cc/2025/Conference — NeurIPS 2025 poster_

### Official Review · Reviewer_W3g8 · 2025-06-20

**Clarity:** 3
**Significance:** 3
**Originality:** 4
**Rating:** 5
**Confidence:** 4

**Summary:**

The authors make two contributions towards the scheduling of a constellation of agile earth observation satellites: 1) a benchmark suite including a simulator which is more detailed and contains more and larger instances, and 2) a novel transformer-based method for finding solutions to this type of scheduling problems. The new approach scores better than the four baselines (inc. random).

**Questions:**

1. In the introduction it says "the operating environment is highly dynamic: new tasks can appear or expire at any moment, satellite positions and attitudes are continuously changing, batteries cycle through charge and discharge, and satellites may even join or leave the constellation. Scheduling algorithms must adapt on the fly without foreknowledge of these changes.". Which of these changes are under control or at least partially influenced by the scheduling algorithm and which uncertainty is completely external?
2. "To bootstrap learning, we curated AEOS-Bench: a large dataset with constellation scheduling annotations." Where does the dataset of tasks come from? And where do the scheduling annotations come from? How has been decided which instances to include and which to exclude?
3. What are the limitations of AEOS-Former? For how many instances does one of the other methods score better? Is there a pattern here, e.g. what properties do those instances have (e.g. considering size)?

**Ethical Concerns:**

["NO or VERY MINOR ethics concerns only"]

**Final Justification:**

In the rebuttal, the authors write that the plan is to include in the revised manuscript:
- a concise discussion of the design alternatives and their rationale
- an analysis to provide a clearer understanding of which instances favor which method
With these additions, I believe this is a good contribution.

For further improvements, the authors could also consider separate experiments to evaluate the learning success with regard to learning physical laws vs dealing with the uncertainty.

**Limitations:**

The limitations are not really discussed nor analyzed. See question 3 above.

**Paper Formatting Concerns:**

- in Figure 5 it says "Constrain Module" instead of "Constraint"
- the space after the , in the numbers seems a bit wide; put in math mode?
- shouldn't w.r.t. be written out?

**Quality:**

3

**Strengths And Weaknesses:**

Strengths:
- two good quality contributions
- the benchmark framework may bring more maturity to the field of multi-satellite scheduling
- the transformer-based approach appears to be novel

Weaknesses:
- the description in the paper provides limited understanding as to why certain design choices of the transformer-based method are taken, and what alternative choices are
- the experiments provide no understanding of which instances work better with which of the baselines, or what the limitations of the transformer-based method are

---

> ### Author Rebuttal · Authors · 2025-07-31
>
> Thank you for the valuable comments and the recognition of our contributions. Below we address specific questions.
>
> # 1. Design Choice of AEOS-Former
> We appreciate your suggestion to better articulate the motivations behind our design choices. At its core, the AEOS scheduling problem asks: "Which satellite should service which task, subject to constraints?" We frame this as a matching problem under constraints, where satellites and tasks are two sets of “tokens” whose pairwise affinities must respect feasibility.
>
> This formulation naturally leads to the choice of **Transformer**, which can ingest variable-sized inputs and excel at reasoning through the attention mechanism. That said, a vanilla Transformer is essentially a black box. Learning to perform task assignments while conforming to all constraints would require massive datasets and prohibitively large compute budgets.
>
> To avoid this, we introduce an **internal constraint module** that explicitly predicts the feasibility of every satellite–task pair. This separation lets the Transformer focus on high-level reasoning, while the lightweight constraint checker guarantees that only physically possible assignments are considered.
>
> Alternative design choices were also considered. A simple **MLP** would fail to handle dynamic input sizes without padding or truncating the inputs. **Graph neural networks** (GNN) over a bipartite satellite-task graph could capture pairwise interactions, but the message-passing schemes would complicate both model design and training, and still leave constraint learning implicit.
>
> By contrast, our matching-based Transformer, augmented with an internal constraint module, strikes the right balance. It naturally accommodates any number of satellites and tasks, leverages powerful attention-driven reasoning, and enforces real-world operational constraints without inflating the training cost. We will include a concise discussion of these design alternatives and our rationale in the revised manuscript.
>
> # 2. Detailed Comparison with Baselines
> Among all baseline methods, MSCPO-SHCS performs closest to our approach. On the **val-unseen** split, it outperforms AEOS-Former on $15$ out of $64$ scenarios. To better understand this, we conducted a statistical analysis comparing the scenarios where each method performs best.
>
> The key differentiating factor appears to be task volume. In scenarios where MSCPO-SHCS outperforms AEOS-Former, the average number of tasks is $168.8$. In contrast, for the remaining scenarios where AEOS-Former performs better, the average number of tasks is $191.3$. This suggests that MSCPO-SHCS is more competitive in smaller-scale settings, likely due to its local search nature being more effective when the solution space is limited. On the other hand, AEOS-Former scales more robustly to high-task-density scenarios, where optimization-based methods tend to struggle with increasing combinatorial complexity.
>
> We will include this analysis in the revised manuscript to provide a clearer understanding of which instances favor which method.
>
> # 3. Limitations
> Due to page limits of the manuscript, we included the limitations section in Sec. 1 of the supplementary materials. In the final manuscript, we will move the limitations section back to the main text.
>
> # 4. Clarification on the Problem Setting
> We appreciate your meticulous review of our manuscript. All of the dynamics mentioned in the introduction (including tasks, satellites, and batteries) are governed by the simulator, instead of the scheduling algorithm. However, not all of these factors are equally uncertain.
> Some properties, such as satellite positions and battery levels, evolve according to physical laws and are therefore predictable given the current state. In contrast, the emergence of new tasks represents true external uncertainty. For example, a task may request observation of a region affected by a sudden natural disaster, something that cannot be predicted in advance. The scheduling algorithm has no prior knowledge of when or where such tasks will be released.
> A central goal of AEOS-Former is to handle this mix of predictable dynamics and unpredictable events in a unified framework. Traditional methods often require re-optimization when the environment changes, which introduces latency. AEOS-Former, by contrast, adapts to all changes without retraining.
>
> # 5. The Data Generation Process
> All tasks in AEOS-Bench are randomly sampled from predefined ranges, as detailed in Tab. 2 of the supplemental material. Each scenario comprises a randomly sampled constellation and task set. Scheduling annotations are then generated using the pipeline described in Fig. 3 of the manuscript: an iterative algorithm proposes candidate schedules, and human reviewers verify their correctness. Only annotations that pass the verification are included in AEOS-Bench.
>
> There are two key stages in our data generation process that involve human oversight. First, during satellite asset generation (illustrated in Fig. 2 of the manuscript), human reviewers filter out satellites with unstable or inefficient attitude control. Only satellites with smooth and reliable pointing behavior are retained as valid assets. Second, during the data annotation phase (as shown in Fig. 3 of the manuscript), human reviewers check for irrational scheduling behavior, such as observations of the same task by multiple satellites.
>
> This pipeline allows us to curate a high-quality dataset of realistic task assignments, while minimizing the burden on human annotators by delegating most of the heavy lifting to automated methods.
>
> # 6. Paper Formatting Concerns
> We appreciate your careful reading and constructive feedback. We will correct the label in Fig. 5 to “Constraint Model.” As for the spacing problem of separators, we have reviewed all mathematical expressions and will address any inconsistent spacing between symbols and numbers during final typesetting. Finally, “w.r.t.” is an abbreviation for “with respect to”, which is widely used in mathematical contexts. We will spell it out to improve readability.

---

> > ### Comment · Reviewer_W3g8 · 2025-08-06
> >
> > Thank you for responding to my review, including the questions asked.
> >
> > 1. It's great to read more about the design alternatives and your rationale; yes, please include this discussion in the paper.
> >
> > 2. The more detailed comparison to the baseline is helpful. Although there is a difference between 168.8 and 191.3 tasks, is it really significant? Doesn't this also depend on the interactions (e.g. closeness) of these tasks? Is that similar? Are there other explanations for the differences?
> >
> > 3. Limitations: in the supplementary only the limitation of not including area tasks in the benchmark and method is mentioned; aren’t there other limitations? E.g. regarding our understanding of the training process (how much is needed?), the success rate of learning the predictable dynamics versus the unpredictable one?
> >
> > 4-6. Thank you; very helpful.

---

> > > ### Author Response · Authors · 2025-08-08
> > >
> > > Thank you for clarifying the questions and highlighting these important points.
> > >
> > > 1. We appreciate the opportunity to improve our manuscript based on your valuable feedback. We will add the discussion to the revised version of our paper.
> > > 2. In our problem setting, the difference between 168.8 and 191.3 tasks (22.5 tasks) accounts for 9.0% of the total range of task counts, which is relatively substantial. Regarding task interactions, **the closeness between tasks changes similarly to the number of tasks.** In scenarios where AEOS-Former outperforms MSCPO-SHCS, the number of neighboring tasks around each task is 14.3% more than the remaining scenarios. **These statistics reinforce our conclusion** that AEOS-Former demonstrates stronger scheduling capability in scenarios with denser tasks.
> > > 3. Unlike prior methods that perform optimization for each scenario individually, AEOS-Former relies on a unified training stage over diverse scenarios. This one-time training cost is necessary to achieve adaptability across different constellations and task distributions, **a trade-off we will discuss in the revised Limitations section.** As for the dynamics, our simulator governs both predictable and unpredictable dynamics as part of our problem setting. AEOS-Former reacts in real-time instead of forecasting the dynamics.
> > >
> > > We hope these detailed clarifications have resolved all ambiguities.

---

### Official Review · Reviewer_Tcxm · 2025-06-30

**Clarity:** 3
**Significance:** 3
**Originality:** 3
**Rating:** 4
**Confidence:** 2

**Summary:**

This work addresses the challenge of scheduling Agile Earth Observation Satellites (AEOSs) under complex, large-scale, and dynamic conditions. The authors introduce AEOS-Bench, the first large-scale, realistic benchmark suite for satellite constellation scheduling, featuring over 16,000 scenarios and 3,900+ satellite assets with ground truth annotations. They also propose AEOS-Former, a Transformer-based model with a constraint-aware attention mechanism and a dedicated module for modeling satellite limitations. AEOS-Former uses simulation-based iterative learning to adapt to diverse scenarios and outperforms existing baselines in both task completion and energy efficiency.

**Questions:**

Who were the experts who annotated the dataset?  Were they subject experts?

What is a scenario? What is an asset?

**Ethical Concerns:**

["NO or VERY MINOR ethics concerns only"]

**Final Justification:**

I have read the rebuttal, and the authors have addressed my concerns and questions. I think the paper should be written more clearly for a better understanding in the next version. The paper makes an important contribution, and it can be accepted.

**Limitations:**

The authors did not discuss any limitations or potential negative societal impact of their work. Can this work be used to create more powerful surveillance systems? How about licensing the code, data or using export controls?

**Quality:**

3

**Strengths And Weaknesses:**

Strengths-

The paper proposes a new scheduling model and a benchmark with 16k scenarios.  The AEOS-Bench and AEOS-Former models are highly useful for real-time, multi-satellite scheduling in Earth observation applications, including disaster response systems.

The solution employs mathematical equations for scenario modeling and multiple constraints to obtain accurate simulation and scheduling. Furthermore, the work introduces a Transformer-based architecture customized with an Internal Constraint Module to predict feasibility and control time, and Cross-attention masking based on learned constraints. The individual ML components (Transformers, iterative learning, attention masking) exist elsewhere. But the cohesive integration of these techniques for the complex, real-world domain of satellite scheduling is interesting.

Good ablation studies.

The performance improvements are marginal when compared to SOTA. The major highlight is the total energy consumed by the satellite sensors during imaging, which is good, as the onboard energy resources on a satellite are a precious commodity.

The writing is good with a clear structure from motivation to results.

The code is provided in the appendix.

Weaknesses -

Some of the concepts are unclear. For example, what is a scenario? What is an asset?

How much energy is saved in doing ML-related tasks on the satellites if the proposed method is used? This is not evaluated.

All loss weights are assigned as ws = wt = wa = 1. Then what is the purpose of these weights in equation (9)? How were these values selected, and what will be the impact on performance if these values are varied? Similarly, how were values for wCR, wPCR, wTAT, etc were selected?

The paper makes two claims: In section 1, it mentions "While effective on simplified benchmarks, their performance degrades sharply in realistic scenarios.", and in Section 2, "While these methods offer faster runtimes, their performance diminishes with large-scale or dynamic scenarios," without backing them up.

There are no details present on tasks like what the tasks were, how they were introduced into the system - randomly, uniformly, what their deadline was.

Minor issues -

Some types, such as out instead of ours.

---

> ### Author Rebuttal · Authors · 2025-07-31
>
> Thank you for your detailed review and insightful questions about our work. Below we address specific questions.
>
> # 1. Concept Clarification
> We apologize for the ambiguity around these terms and appreciate the opportunity to clarify. In our work, a scenario refers to a scheduling problem instance at timestep $0$, which comprises the constellation state (positions, attitudes, remaining resources) and a set of tasks (target coordinates, time windows). During inference, the AEOS-Bench simulator first loads the scenario to initialize the constellation, then steps through time. At each subsequent timestep, the simulator applies the high‑level scheduling commands from AEOS‑Former, updates the constellation state via the MRP controller, and records task progress until the scenario concludes.
>
> An asset denotes a satellite eligible for inclusion in AEOS-Bench. As shown in Fig. 2 of the manuscript, we generate a large pool of assets by sampling satellite properties, computing corresponding MRP parameters via an empirical formula, and then filtering through a completion rate check and a human quality check. The surviving satellites become part of the asset pool. When constructing scenarios for AEOS-Bench, constellations are formed by random sampling from the asset pool.
>
> We will incorporate these definitions of "scenario" and "asset" into the revised manuscript to eliminate any remaining confusion.
>
> # 2. Power Consumption
> We presume the "energy" is referring to the **onboard power consumption** of satellites during task execution. As shown in Tab. 2 of the manuscript, AEOS-Former achieves a significant reduction in power usage compared to baseline methods. For example, on the **val-unseen** split of AEOS-Bench, our method reduces power consumption from **140.83 Wh** (MSCPO-SHCS) to **68.99 Wh**, representing an approximate 50% reduction.
>
> Although satellites are typically equipped with solar panels for recharging, minimizing power consumption has clear benefits: it enables more flexible maneuver and reduces wear on high-power components—ultimately helping to extend satellite lifespan without requiring additional hardware robustness.
>
> # 3. Ablation Study on Loss Weights
> The loss weights $(w_s, w_t, w_a)$ in Eq. (9) are designed to balance the contributions of feasibility loss, timing loss, and task assignment loss. We selected $ws=wt=wa=1$ based on preliminary experiments, where an ablation study (summarized in the table) showed that AEOS-Former is robust to the loss weights and setting $w_s = w_t = w_a = 1$ achieves **better performance**, which is shown below:
>
> | $w_s$ | $w_t$ | $w_a$ | CS $\downarrow$ | CR/% | PCR/% | WCR/% | TAT/h $\downarrow$ | PC/Wh $\downarrow$ |
> |---|---|---|---|---|---|---|---|---|
> | 1.0 | 1.0 | 1.0 | **4.43** | **35.42** | **38.93** | **35.14** | 6.78 | 68.99 |
> | 0.5 | 1.0 | 1.0 | 4.74 | 31.71 | 34.57 | 31.65 | 6.91 | 65.31 |
> | 1.5 | 1.0 | 1.0 | 4.79 | 31.41 | 34.26 | 31.37 | 7.05 | 65.58 |
> | 1.0 | 0.5 | 1.0 | 4.65 | 32.47 | 35.38 | 32.40 | 6.75 | 66.17 |
> | 1.0 | 1.5 | 1.0 | 4.68 | 32.03 | 34.75 | 31.97 | 6.67 | 65.79 |
> | 1.0 | 1.0 | 0.5 | 4.84 | 30.55 | 33.30 | 30.82 | 6.83 | **65.02** |
> | 1.0 | 1.0 | 1.5 | 5.49 | 24.58 | 27.37 | 24.53 | **5.82** | 67.48 |
>
> We did not perform an extensive hyperparameter sweep over the three loss weights. Thus, in future work, we plan to conduct more systematic hyperparameter searches to determine whether alternative weight combinations can yield further improvements.
>
> # 4. Rationale of the CS Definition
> We define CS as a **comprehensive metric** that balances completion rate metrics (CR, PCR, WCR) with efficiency metrics (TAT, PC). To do so, we first aggregate the three completion rate metrics into a single completion ratio, and then combine this with normalized versions of TAT and PC.
>
> Specifically, we compute the completion ratio as a weighted average of CR, PCR, and WCR. The weights ($w_\text{CR} = 0.6, w_\text{PCR} = 0.2, w_\text{WCR} = 0.2$) reflect our prioritization of CR over PCR and WCR. This aggregated ratio naturally lies in $[0, 1]$, where higher is better. In contrast, TAT and PC are unbounded positive quantities that we wish to minimize. To align all components, we take the reciprocal of the completion ratio, mapping it to $[1, \infty)$.
>
> Then we normalize TAT and PC with their respective normalization weights. To choose the values for $w_\text{TAT}$ and $w_\text{PC}$, we examined the typical ranges of these metrics. We observed that TAT hovers around $7$ hours and PC around $100$ Wh. Setting $w_\text{TAT} = 1/7$ and $w_\text{PC} = 1/100$ therefore brings these terms into the same order of magnitude (around $1$), so that no single metric overwhelms the overall score. Random, HAAL, and REDA were omitted from this calibration because their completion rate metrics were anomalously low and would skew the normalization.
>
> This formulation ensures that CS accounts **fairly** for both coverage performance and resource efficiency, with no undue bias toward any single metric.
>
> # 5. Performance of Prior Works
> Thank you for pointing out the need for concrete evidence behind these two claims.
>
> First, regarding "While effective on simplified benchmarks, their performance degrades sharply in realistic scenarios," Tab. 2 of our manuscript shows that REDA achieves 21.54 CS on the val‑unseen split, whereas AEOS‑Former achieves 4.43 CS. On the original REDA benchmark (which ignores attitude control, sensor FOV, etc.), REDA and AEOS‑Former score $11,094$ and $17,325$ in total return, respectively. Although REDA still falls short there, the performance gap shrinks on the simplified benchmark. This contrast confirms that the performance of REDA degrades much more severely when faced with the realistic dynamics and constraints embedded in AEOS‑Bench.
>
> Second, for "While these methods offer faster runtimes, their performance diminishes with large‑scale or dynamic scenarios," we performed a scenario‑level analysis comparing MSCPO‑SHCS and AEOS‑Former. We found that MSCPO‑SHCS outperforms AEOS‑Former only on scenarios with an average of 168.8 tasks, whereas AEOS‑Former takes the lead on scenarios averaging 191.3 tasks. This pattern demonstrates that while MSCPO‑SHCS may be competitive on smaller task sets, AEOS‑Former’s learning‑based approach scales gracefully to higher task densities.
>
> # 6. Task Definition
> Tasks in AEOS-Bench are defined as satellite observation missions, each characterized by key parameters: minimum continuous observation duration, release time, due time, and target coordinates. In AEOS-Bench, these parameters are sampled independently and uniformly from predefined ranges, as fully specified in Tab. 2 of the supplemental material.
>
> # 7. Typos
> Thank you very much for pointing out our typo. We will strive for excellence and correct all typos in the revised manuscript.
>
> # 8. Data Annotation
> Annotating ground‑truth schedules by hand is laborious. Annotating a single scenario with $10$ satellites and $50$ tasks can take an expert nearly $40$ minutes. To streamline this, we developed the pipeline in Fig. 3 of the manuscript: an iterative algorithm proposes candidate schedules, and human reviewers simply verify them rather than building them from scratch. Because reviewers only need to confirm rationality, this task requires **minimal** specialized astronautics knowledge.
>
> All human validation was performed by the paper’s authors. In future work, we plan to expand our annotation effort via crowdsourcing to scale AEOS‑Bench even further while maintaining high quality.
>
> # 9. Limitations and Broader Impacts
> Due to page limits of the manuscript, we provide detailed discussions of limitations (Sec. 1) and broader impacts (Sec. 2) in our supplementary materials.
>
> Our work is designed with socially beneficial applications in mind, such as ecological monitoring and disaster response of AEOS constellations. While it is possible that the scheduling technology could be applied in surveillance contexts, we strongly oppose any misuse. Importantly, the **Outer Space Treaty** adopted by the United Nations prohibits harmful interference and hostile activities in space, including unlawful surveillance, providing a legal framework that governs responsible use of space technologies worldwide.
>
> To further mitigate risks, our source code is licensed under Apache 2.0, but we plan to introduce additional restrictions, including provisions that explicitly limit downstream misuse. For AEOS-Bench, access will require submission of an application form and approval by the authors. Redistribution of the dataset will be strictly prohibited without explicit permission.
>
> We take the potential for negative societal impact seriously and are actively incorporating safeguards into both our licensing and data access policies to ensure that our work is used ethically and in accordance with international norms.

---

### Official Review · Reviewer_fTGm · 2025-07-02

**Clarity:** 2
**Significance:** 3
**Originality:** 3
**Rating:** 4
**Confidence:** 3

**Summary:**

This paper addresses the scheduling challenges faced by Agile Earth observation satellite (AEOS) constellations in large-scale and dynamic scenarios.
The authors propose a dual innovation strategy by integrating a standardized benchmark and a novel scheduling model to alleviate these challenges.
The key contributions are as follows:
1. AEOS-Bench Benchmark:
The authors introduce AEOS-Bench, the first large-scale, high-fidelity scheduling benchmark consisting of 3,907 satellite assets and 16,410 dynamic scenarios (ranging from 1–50 satellites and 50–300 tasks). The dataset is generated using a physics-based simulation platform (Basilisk engine), providing realistic orbital dynamics and energy constraints along with labeled scheduling data.
2. AEOS-Former Scheduling Model:
A Transformer-based model, AEOS-Former, is proposed with a novel internal constraint module. This module explicitly encodes physical limitations of satellites (e.g., sensor field of view, battery status), and uses a constraint-driven attention mechanism to generate feasible scheduling policies. Further, the model is trained using a hybrid learning framework that combines supervised pre-training with simulation-based self-optimization, leading to significant improvements in task completion rate and energy efficiency.

**Questions:**

**Questions**

- The paper does not elaborate on the benchmark’s use cases in sufficient detail. Could the authors clarify their choice of track, and explain why the benchmark/dataset track was not considered more appropriate for this submission?

**Suggestions**

- Generalization to Unseen Scenarios: While the model performs reasonably well under the val-unseen split setting, its completion rate and partial completion rate on the test split fall below those of some baseline methods. This suggests that the model’s generalization to truly unseen scenarios remains limited. The authors are encouraged to further analyze this performance gap and consider possible enhancements to improve adaptability in novel or out-of-distribution environments.
- Scalability to Large-Scale Constellations: While the current setup supports up to 50 satellites, many operational constellations (e.g., Starlink, OneWeb) involve hundreds or thousands of satellites. The authors are encouraged to strengthen their evaluation or at least discuss the potential scalability of their approach under such large-scale constellation settings.
- Impact of High-Level Scheduling Decoupled from Low-Level Control: The proposed method generating high-level scheduling commands, which are then translated into low-level control actions (e.g., attitude maneuvers) by downstream systems. However, it remains unclear whether this decoupling between high-level scheduling and low-level execution might introduce inaccuracies or latency during real-world operations. A brief discussion on this potential issue—such as how the system ensures feasibility and timing alignment—would strengthen the clarity and completeness of the system design.
- Temporal Granularity and Feature Fusion:
The model jointly embeds static orbital parameters and dynamic satellite states (e.g., real-time attitude). However, orbital dynamics typically evolve over hour-level timescales, whereas attitude control changes on the order of seconds. This significant mismatch in temporal granularity might require a hierarchical or decoupled processing scheme. The authors are encouraged to discuss whether such a temporal difference affects decision-making and how it is handled in the current model.

**Ethical Concerns:**

["NO or VERY MINOR ethics concerns only"]

**Final Justification:**

The authors' detailed response has addressed most of my concerns. It may be worthwhile to explore scalable system designs that can support such expanded deployments.

**Limitations:**

- Lightweight Deployment on Resource-Constrained Platforms.  Given that satellites typically operate in resource-constrained environments, it would be valuable for the authors to explore lightweight inference or training strategies suitable for deployment on onboard SoC (COTS) platforms. Discussing techniques such as model pruning, quantization, or knowledge distillation would enhance the practical applicability of the proposed method (in the future work).

**Quality:**

3

**Strengths And Weaknesses:**

**Strengths**

- High-Fidelity Benchmark Construction:
The authors leverage a physics-based simulation engine to model orbital dynamics, energy cycles, and other physical constraints with expert-reviewed realism. The benchmark scenarios are diverse, covering a wide range of orbital parameters and task loads.

- Scalability and Robustness:
The proposed model supports scheduling for 1–50 satellites and demonstrates strong robustness in the presence of dynamic task arrivals and fluctuating onboard resources. This reflects good scalability toward real-world AEOS operations.

- Integration of Learning and Constraint Modeling:
The work tightly integrates benchmark construction, explicit constraint modeling, and reinforcement learning. The AEOS-Former achieves superior performance across multiple metrics compared to baseline approaches.

**Weaknesses and concerns**
- High Computational Cost:
Training AEOS-Former requires 8 RTX 4090 GPUs and nearly 48 GPU-hours, which raises concerns about computational efficiency. This cost may hinder real-time deployment in resource-constrained onboard environments.

- Limited Validation on Truly Large-Scale Constellations:
While the model supports up to 50 satellites, many real-world constellations involve significantly more satellites (e.g., hundreds or thousands). It would strengthen the work to evaluate or at least discuss scalability and limitations in truly large-scale constellations.

- Execution Interface and Control Fidelity:
The current model outputs high-level scheduling commands, which still depend on downstream attitude control systems for actual execution. This raises a question: could the decoupling from low-level control introduce execution inaccuracies or delays?

---

> ### Author Rebuttal · Authors · 2025-07-31
>
> We appreciate your detailed review and constructive feedback very much. We provide our response to your review as follows.
>
> # 1. Computation Cost and Deployment
> Although our model requires $6$ hours for training completion (with $8$ RTX4090 GPUs), the training cost is incurred only once. Upon training completion, the model can **directly** perform inference on new scenarios. Consequently, deployment in real-world applications does not necessitate substantial computational resources for retraining, but rather enables deployment through invocation of the pre-trained model. Moreover, **this training cost can be further reduced**. Our $6$-hour training duration is implemented to achieve superior performance. As demonstrated in the table below, the model exhibits satisfactory convergence after $2$ hours of training on $8$ RTX4090 GPUs, achieving a CS score of $4.67$.
>
> | GPU-hours | CS $\downarrow$ | CR/% | PCR/% | WCR/% | TAT/h $\downarrow$ | PC/Wh $\downarrow$ |
> |---|---|---|---|---|---|---|
> | 16 (30k iters) | 4.67 | **38.17** | **41.52** | **38.22** | 7.82 | 97.68 |
> | 32 (60k iters) | 4.55 | 37.62 | 41.13 | 37.31 | 7.82 | 82.34 |
> | 48 (90k iters) | **4.43** | 35.42 | 38.93 | 35.14 | **6.78** | **68.99** |
>
> Regarding deployment challenges under resource-constrained conditions, as illustrated in the table below, the model's required FLOPS do not operate at an excessively high level. Taking the RK3568, a commonly employed Commercial Off-The-Shelf (COTS) platform for satellite deployment, as an example, its FP32 computational capacity approximates $500$ GFLOPS, which is sufficient to achieve second-level task allocation for the majority of scenarios presented in the table. Since constellation scheduling constitutes a high-level task, frequent execution at millisecond granularity does not impact allocation outcomes but with higher power consumption. Therefore, **second-level allocation is entirely adequate for constellation mission planning applications**. That said, we agree that model compression techniques can further boost the performance. In future work, we will implement model quantization and pruning to better accommodate the demands of larger-scale constellation planning over the coming decades.
>
> | #Satellites | 10 | 50 | 100 | 1000 | 50 | 50 | 50 |
> |---|---|---|---|---|---|---|---|
> | #Tasks | 300 | 300 | 300 | 300 | 50 | 100 | 1000 |
> | GFLOPS | 54.29 | 165.25 | 303.94 | 2800.50 | 31.24 | 58.04 | 540.47 |
> | Latency (ms) | 1.376 | 1.796 | 2.286 | 11.577 | 1.385 | 1.454 | 2.878 |
>
>
> # 2. Scalability to Large-Scale Constellations
> As mentioned in the table below, this model can be applied to larger-scale constellations without additional training and can essentially cover larger-scale real-world constellations. Regarding satellite internet constellations such as Starlink, although they contain massive constellations (typically thousands of satellites in different orbits), they differ fundamentally from the remote sensing satellites targeted in this paper: internet constellations focus on providing continuous services to broader regions and generally do not concern themselves with signal directionality, meaning they only need to consider satellite coverage of ground areas. In contrast, remote sensing satellites face extensive remote sensing demands across various regions and require specialized scheduling for specific locations. Moreover, due to the difficulty in miniaturizing the optical equipment they carry, it is challenging to launch multiple satellites simultaneously. This indirectly results in remote sensing constellations being at most in the hundreds scale, even when not small in size. Most remote sensing constellations planned for the next decade are around the scale of $100$ satellites. Therefore, this model possesses good scalability for constellations that require constellation planning of this type.
>
> | #Satellites | #Tasks | CS $\downarrow$ | CR/% | PCR/% | WCR/% | TAT/h $\downarrow$ | PC/Wh $\downarrow$ |
> |---|---|---|---|---|---|---|---|
> | 1-50 | 50-300 | 4.43 | 35.42 | 38.93 | 35.14 | 6.78 | 68.99 |
> | 50-100 | 100-300 | 5.38 | 64.71 | 69.73 | 64.51 | 12.14 | 212.13 |
> | 1-50 | 100-600 | 6.62 | 24.30 | 27.69 | 23.93 | 11.98 | 89.38 |
>
> # 3. Accuracy and Timeliness of Low-Level Attitude Control
> We appreciate your insight regarding the low‑level attitude control. In our framework, AEOS‑Former outputs only the high-level task assignments (specifying which satellite observes which target) at each timestep, while the AEOS-Bench simulation handles low-level attitude control for each satellite. As detailed in Fig. 1 of the supplemental materials, the simulator first computes the attitude tracking error (the angular separation between the satellite's current attitude and the task's line of sight). Then, the Modified Rodrigues Parameters (MRP) feedback control algorithm [1] adjusts the angular velocities of reaction wheels to control the attitude. MRP, much like a PID controller but tailored for spacecraft attitude, is carefully tuned to optimize operational accuracy and timeliness. As long as the high‑level commands generated by AEOS‑Former respect the constraints of the satellite, the MRP controller can execute these commands without introducing additional latency or inaccuracy.
>
> This decoupled architecture, where a task planner issues high-level commands and a specialized controller executes them, is widely used in Embodied AI and robotics [2, 3]. This separation allows the planner to operate at a lower frequency that is sufficient for strategic scheduling, while the low‑level controller runs at high rates to ensure stable and precise attitude control. **By delegating low-level control to MRP, AEOS‑Former can focus on optimizing task assignments, which leads to better scheduling performance and more accurate evaluation.**
>
> We agree that an end‑to‑end co‑optimization of scheduling and control could yield further gains, as the model would have direct influence over reaction wheels. Investigating such integrated approaches, potentially via a hierarchical design, is an exciting direction for future work.
>
> [1] Analytical Mechanics of Space Systems. AIAA. DOI: 10.2514/4.102400.
>
> [2] Helix: A Vision-Language-Action Model for Generalist Humanoid Control. Figure.
>
> [3] Agent as cerebrum, controller as cerebellum: Implementing an embodied lmm-based agent on drones. arXiv: 2311.15033.
>
> # 4. Choice of Track
> We carefully considered submitting to the dataset/benchmark track before deciding on the main track. Our primary goal was to present a complete solution to the realistic AEOS constellation scheduling problem, one that not only provides a realistic benchmark but also introduces a novel constraint‑aware Transformer architecture and training pipeline.
>
> Existing benchmarks either focus on toy scenarios or omit critical constraints, failing to support the training and rigorous evaluation of scheduling models for real-world deployment. Therefore, we develop AEOS-Bench as an essential component for solving the realistic AEOS constellation scheduling problem, though it is not an end in itself. While we believe AEOS‑Bench will be a valuable resource for the community, our contribution also lies in the methodological innovations of AEOS-Former.
>
> Submitting to the main track allows us to present the **full narrative**: from problem formulation to network design and training. We believe this integrated presentation better reflects the breadth and depth of our contributions and will more effectively inspire future work.
>
> # 5. Completion Rate on the Test Split
> The reduced CR metric on the test split is primarily driven by its internal constraint module $\mathcal{C}$, which filters out satellite-task pairs deemed infeasible. On both the val-seen and val-unseen splits, the PC metric remains around $70$ Wh, but on the test split, PC falls to $40.91$ Wh, indicating that $\mathcal{C}$ rejects a substantially larger fraction of assignments. The underlying cause is a sim-to-real gap. The satellite parameters in AEOS-Bench are uniformly sampled (Tab. 1 of the supplemental material), while real-world satellite parameters often follow different distributions. As a result, the feasibility checks of $\mathcal{C}$ are overly conservative on the test split.
>
> An immediate remedy is to adjust the feasibility threshold $\tau$ of $\mathcal{C}$. In Tab. 3 of the supplemental material, lowering $\tau$ from $1 \times 10^{-3}$ to $5 \times 10^{-4}$ increases test-split CR from $19.25$ to $20.37$, surpassing the MSCPO-SHCS baseline ($19.44$ CR) and reduces CS from $6.35$ to $6.18$. This simple tuning **recovers** much of the lost performance.
>
> # 6. Temporal Granularity
> In astronautics, the orbital dynamics of a satellite are affected by the orbital altitude. Our research focuses on AEOS satellites, which typically operate in Low-Earth Orbit (LEO), with altitudes ranging from $200$ km to $2000$ km. At these altitudes, LEO satellites travel at approximately 7.9 km/s and complete an orbit around the Earth in merely $90$-$120$ minutes. As a result, their orbital state evolves significantly in the order of seconds. This temporal granularity aligns closely with that of attitude control.
>
> At each timestep, we feed the full satellite state (including both orbital parameters and attitude parameters) into AEOS-Former to make task assignments.

---

> > ### Comment · Reviewer_fTGm · 2025-08-01
> >
> > The authors' detailed response has addressed most of my concerns. However, I still have some reservations regarding one particular viewpoint: the authors argue that the remote sensing satellite constellations proposed in this work are fundamentally different from satellite internet constellations such as Starlink, and thus predict that the scale of such remote sensing constellations will remain within hundreds of satellites in the coming decades.
> >
> > Nevertheless, recent developments in satellite internet constellations suggest a clear trend toward the convergence of communication, remote sensing, and navigation functionalities within integrated satellite architectures. This evolution indicates the potential for larger-scale satellite constellations in the future. Therefore, it may be worthwhile to explore scalable system designs that can support such expanded deployments.

---

> > > ### Author Response · Authors · 2025-08-02
> > >
> > > We appreciate your quick feedback and the **insightful predictions for next-generation constellations**. Characterized by large scale and multifunctionality, these constellations will undoubtedly demand more sophisticated scheduling solutions. Our experiments show that AEOS-Former generalizes well beyond its training scale, maintaining performance as constellation size grows. We leave the extension to next-generation constellations for future work, which will require substantial enhancements. In future work, the next-generation constellation scenarios will be included in AEOS-Bench for training and evaluation at the next level of scale. To further enable AI-driven coordination across communication, navigation, and sensing functions, a potential future work is to develop unified feature representations for heterogeneous payloads. **Such future works and research directions will mark a significant step toward versatile scheduling for next-generation constellations.**

---

### Official Review · Reviewer_FRRA · 2025-07-03

**Clarity:** 2
**Significance:** 2
**Originality:** 3
**Rating:** 5
**Confidence:** 1

**Summary:**

The authors tackle the problem of large-scale satellite constellation scheduling. They introduce a new benchmark (AEOS-Bench) obtained through high-fidelity simulations. Further, a new architecture is proposed (AEOS-Former) which is a Transformer-based scheduling model featuring a constraint-aware attention mechanism and a dedicated internal constraint module that explicitly models the physical and operational limits of each satellite. This model is compared on the proposed benchmark to various baselines and demonstrates superior performance.

**Questions:**

* Are all of the presented baselines neural network based?
* Were also Optimization-based Methods evaluated on the benchmark?
* How do you make sure that the sequences from the simulation in the data gathering phase are not present in the evaluation phase as it uses the same simulator?

**Ethical Concerns:**

["NO or VERY MINOR ethics concerns only"]

**Final Justification:**

The authors addressed all my questions and concerns by their response.

**Limitations:**

No Limitations section and not really discussed in the Conclusion.

**Paper Formatting Concerns:**

No concerns

**Quality:**

3

**Strengths And Weaknesses:**

**Strengths**

* The paper is well written and the methods are explained comprehensively.
* The introduced benchmark is a nice contribution to the community surpassing earlier benchmarks quite significantly by e.g. number of scenes (Table 1).
* The authors provide an extensive ablation study on the individual components of AEOS-Former.


**Weaknesses**

* Experiments are only performed on the introduced new benchmark. It would be helpful to also evaluate the model on some of the other benchmarks presented in Table 1 to get an understanding whether the method is specially tuned for this particular benchmark.
* A bit more context on the different baselines would have made it easier to understand the improvement and how the method compares to earlier methods.

---

> ### Author Rebuttal · Authors · 2025-07-31
>
> Thank you for your meticulous review and constructive comments. Below we address specific questions.
>
> # 1. Experiments on Previous Benchmarks
> We agree that testing AEOS-Former on external benchmarks is important to demonstrate its broader applicability. To that end, we directly evaluated our pretrained AEOS-Former on the REDA benchmark, which comprises a single scenario with $324$ satellites and $450$ tasks. Unlike AEOS-Bench, REDA does not simulate attitude dynamics. Instead, it measures reward solely based on the proximity between satellites and task locations. Since AEOS-Former relies on attitude information as part of its input, we use the AEOS-Bench simulator to generate the necessary attitude information for each REDA satellite.
>
> Introducing these attitude constraints naturally prevents AEOS-Former from selecting some task assignments that would maximize the reward metric. Nevertheless, even under this more restrictive regime, AEOS-Former achieved a total reward of $17,325$, outperforming REDA’s score of $11,094$ by a wide margin.
>
> This result shows that AEOS-Former is not over-fitted to AEOS-Bench but can also generalize to much larger constellations and different evaluation criteria. We will include these cross-benchmark results in the revised manuscript to underscore the versatility and robustness of our approach.
>
> # 2. Context on Baselines
> The baselines presented in our paper include both optimization-based and learning-based methods. Specifically, **HAAL** and **MSCPO-SHCS** are optimization-based approaches. HAAL formulates the scheduling problem as an integer-constrained optimization task, solving it using traditional planning solvers. MSCPO-SHCS adopts a stochastic hill-climbing strategy to iteratively refine scheduling decisions in a time-efficient manner. In contrast, **REDA** is a neural network-based method that combines multi-agent reinforcement learning with polynomial-time greedy solvers to strike a balance between solution quality and computational speed.
>
> As shown in Table 2 of the manuscript, we evaluated all of these methods on AEOS-Bench. Our AEOS-Former demonstrates consistent improvements over both optimization-based and learning-based baselines. We will expand the background and clarify the distinctions between baseline methods in the revised version to improve readability and context.
>
> # 3. Uniqueness of Scenarios
> Every scenario in AEOS-Bench is guaranteed to be unique. Even across different splits, no two scenarios share the same constellation or task set. During simulation, the AEOS-Bench simulator first loads the scenario to initialize the constellation, then steps through time. At each subsequent timestep, the simulator applies the high‑level scheduling commands, updates the constellation state, and records task progress until the scenario concludes. When gathering training data and when running evaluations, the simulator is indeed the same, but the scenarios are always different. As the simulator steps through time, it produces a distinct trajectory of constellation states and task progress for each scenario.
>
> Since the train, val-seen, val-unseen, and test splits are disjoint pools of scenarios, no exact sequences from data collection appear during evaluation. This prevents the model from memorizing specific simulation runs, forcing it to learn scheduling strategies that generalize to **entirely new** constellations and task sets.
>
> # 4. The Limitations Section
> A dedicated limitations section in the main text is omitted due to page limits. The discussion is included in Sec. 1 of the supplemental material. In the final manuscript, we will move the limitations section back to the main text.

---

> > ### Comment · Reviewer_FRRA · 2025-08-07
> >
> > Thank you for addressing my open points and the clarifications. I raised my score to an accept.

---

> > > ### Author Response · Authors · 2025-08-09
> > >
> > > Thanks for your recognition of our work and the valuable comments. We are glad that all your concerns are addressed. We will refine our manuscript to include the experiments on previous benchmarks and richer context as you pointed out.

---

### Official Review · Reviewer_ACJP · 2025-07-07

**Clarity:** 3
**Significance:** 3
**Originality:** 3
**Rating:** 5
**Confidence:** 4

**Summary:**

The paper presents a comprehensive framework for scheduling constellations of Agile Earth Observation Satellites (AEOSs), addressing the challenge of efficiently assigning imaging tasks to multiple satellites under dynamic conditions and stringent constraints. The authors introduce AEOS-Bench, a large-scale benchmark suite containing 3,907 satellite assets and 16,410 scenarios, generated through high-fidelity simulations to ensure realistic satellite behavior and operational constraints. Each scenario includes ground truth scheduling annotations. Building on this benchmark, they propose AEOS-Former, a Transformer-based scheduling model with a constraint-aware attention mechanism. This model incorporates an internal constraint module that explicitly models each satellite's physical and operational limits, such as battery state and sensor field of view. AEOS-Former is trained through a simulation-based iterative learning process, adapting to diverse scenarios and improving scheduling robustness.

**Questions:**

1. How to extend the proposed solution to new constellations and tasks?
2. How to quantify the computational cost?
3. Any optimality or convergence properties of the proposed model?

**Ethical Concerns:**

["NO or VERY MINOR ethics concerns only"]

**Final Justification:**

Based on the author rebuttal for all reviewer comments, I think they further clarified my previous concerns and I would vote for an acceptance.

**Limitations:**

yes

**Paper Formatting Concerns:**

No concerns

**Quality:**

3

**Strengths And Weaknesses:**

# Strengths
1. The paper introduces AEOS-Bench, a large-scale benchmark suite with 16,410 scenarios and 3,907 satellite assets, generated using a high-fidelity simulation platform. This ensures realistic satellite behavior, including orbital dynamics and resource constraints. The inclusion of ground truth scheduling annotations further enhances its utility.
2. The benchmark is diverse, covering a wide range of satellite and task configurations (1 to 50 satellites and 50 to 300 tasks). It is publicly accessible, allowing researchers to evaluate and compare different scheduling models under realistic conditions.
3. The use of a Transformer-based architecture allows the model to effectively capture contextual relationships between satellites and tasks. The constraint-driven attention mechanism guides the scheduling process, improving both efficiency and feasibility.
# Weaknesses
1. The training process of AEOS-Former requires significant computational resources (approximately 48 GPU-hours).
2. While the model demonstrates strong performance on the provided benchmark, its scalability to even larger constellations or more complex scenarios is not fully explored. The computational cost may increase prohibitively with larger problem sizes.
3. The Transformer-based architecture, while powerful, may lack interpretability compared to simpler heuristic or optimization-based methods.
4. The paper does not provide theoretical guarantees or proofs regarding the optimality or convergence properties of the proposed model.

---

> ### Author Rebuttal · Authors · 2025-07-31
>
> Thank you for your thoughtful feedback and the recognition of our work. Below we address specific questions.
>
> # 1. Computation Cost for Training
> While AEOS‑Former requires approximately 48 GPU-hours to train ($6$ hours on $8$ RTX 4090 GPUs), we believe this cost is justified given the complexity of the AEOS constellation scheduling task and the significant generalization benefits it enables.
> Following the scaling law of Transformers [1], model performance improves with increased model size, dataset size, and training cost. In the AEOS constellation scheduling problem, inference latency is strictly constrained, thus limiting the model size. Therefore, **we scale up the dataset size to cover a wide diversity of scenarios and prolong the training process to better optimize the model.**
>
> While the training cost is more expensive than some heuristic or optimization-based methods, it is a **one-time cost**. The resulting model exhibits **robust generalization to unseen or changing scenarios** (e.g., a satellite malfunctions) without retraining, enabling real-time scheduling (latency in milliseconds). In contrast, prior methods are often tailored to fixed scenarios; when the scenario changes, re-running the optimization procedures (taking minutes to hours) is required, which severely hinders real-time responsiveness.
>
> To further investigate the trade-off between training cost and performance, we conducted a small-scale ablation. As summarized in the table below, **AEOS‑Former retains high performance even under reduced training budgets**:
>
> | GPU-hours | CS $\downarrow$ | CR/% | PCR/% | WCR/% | TAT/h $\downarrow$ | PC/Wh $\downarrow$ |
> |---|---|---|---|---|---|---|
> | 16 (30k iters) | 4.67 | **38.17** | **41.52** | **38.22** | 7.82 | 97.68 |
> | 32 (60k iters) | 4.55 | 37.62 | 41.13 | 37.31 | 7.82 | 82.34 |
> | 48 (90k iters) | **4.43** | 35.42 | 38.93 | 35.14 | **6.78** | **68.99** |
>
> To lower the computational barrier for the community, we will release both the AEOS-Bench dataset and the pretrained AEOS-Former model. This allows researchers to finetune or directly deploy the model, mitigating the need for expensive pretraining.
>
> [1] Scaling laws for neural language models. arXiv: 2001.08361.
>
> # 2. Scalability to More Complex Scenarios
> Since AEOS‑Former formulates scheduling as a matching problem between satellites and tasks, it extends seamlessly to arbitrary new constellations and task sets. Currently, each scenario in AEOS-Bench contains $1$-$50$ satellites and $50$-$300$ tasks. To test the scalability of AEOS-Former, we evaluate the pretrained model on more complex scenarios that lie outside the original range of AEOS-Bench. The results are summarized below:
>
> | #Satellites | #Tasks | CS $\downarrow$ | CR/% | PCR/% | WCR/% | TAT/h $\downarrow$ | PC/Wh $\downarrow$ |
> |---|---|---|---|---|---|---|---|
> | 1-50 | 50-300 | 4.43 | 35.42 | 38.93 | 35.14 | 6.78 | 68.99 |
> | 50-100 | 100-300 | 5.38 | 64.71 | 69.73 | 64.51 | 12.14 | 212.13 |
> | 1-50 | 100-600 | 6.62 | 24.30 | 27.69 | 23.93 | 11.98 | 89.38 |
>
> When we increase the number of satellites to $50$-$100$, CR jumps from $35.42$ to $64.71$, indicating that AEOS‑Former effectively leverages additional resources without retraining. With more tasks ($100$-$600$), CR drops to $24.30$ since the task load exceeds the satellite resources. These experiments demonstrate that AEOS-Former retains strong generalization when facing complex scenarios **beyond its training distribution**.
>
> # 3. Computation Cost for Inference
> We sincerely appreciate you for emphasizing this critical concern about computational scalability. We have quantified the computational cost through two key metrics: arithmetic complexity (GFLOPS) and inference latency (to directly reflect real-time feasibility).
>
> We use the `calflops` tool to measure the GFLOPS across varying scenario sizes (the number of satellites and tasks). While Transformer modules theoretically exhibit quadratic complexity, it contributes only $10%$ in the total GFLOPS of AEOS-Former. The remaining $90%$ stems from our constraint module, which scales linearly with the scenario size. As shown below, the GFLOPS of AEOS-Former grows linearly with the scenario size:
>
> | #Satellites | 10 | 50 | 100 | 1000 | 50 | 50 | 50 |
> |---|---|---|---|---|---|---|---|
> | #Tasks | 300 | 300 | 300 | 300 | 50 | 100 | 1000 |
> | GFLOPS | 54.29 | 165.25 | 303.94 | 2800.50 | 31.24 | 58.04 | 540.47 |
> | Latency (ms) | 1.376 | 1.796 | 2.286 | 11.577 | 1.385 | 1.454 | 2.878 |
>
> We benchmark the inference latency over $1,000$ runs on an RTX 4090 GPU to assess real-world performance. Even at the largest scale ($1,000$ satellites + $300$ tasks), AEOS‑Former completes in $11.58$ ms. For smaller scenarios (e.g., $100$ satellites + $100$ tasks), latency drops to under $3$ ms. Since our scheduling system operates at a $1$ Hz update rate, these latencies **leave ample margin** for real‐time operation.
>
> # 4. Interpretability of AEOS-Former
> Traditional heuristic or optimization-based schedulers provide full transparency by applying explicit decision rules or optimizing known objective functions at each step. In contrast, neural network (NN) models, including AEOS-Former, are inherently "black boxes". It is challenging to trace exactly how individual scheduling decisions arise. This lack of transparency is a general limitation of NN-based methods.
>
> To improve interpretability, we introduce an internal constraint module that explicitly predicts the feasibility of each satellite-task pairing. By exposing these feasibility scores, users can observe which assignments are permitted or filtered out. While this does not fully reveal the model’s internal reasoning, it introduces a verifiable layer that partially bridges the interpretability gap between AEOS-Former and traditional heuristic or optimization-based approaches.
>
> # 5. Convergence of AEOS-Former
> Optimization-based methods perform iterative optimization for specific scenarios, attempting to find optimal solutions. As the number of iterations increases, the probability of finding the optimal solution becomes higher. Our method directly takes scenario information as input and outputs task assignment without any iterative process. Therefore, the performance of our method primarily depends on the degree of fitting to the training set and the generalization error. Quantitative research on the fitting degree and generalization error of deep neural networks is still in a relatively foundational stage. Some studies have achieved global convergence of encoder-only shallow transformers [1]. However, we acknowledge that the performance of deep neural networks, compared to optimization-based methods, lacks theoretical guarantees and mainly relies on statistical results on test data to qualitatively demonstrate effectiveness. In future work, we will attempt to integrate deep neural networks with optimization-based methods to achieve stronger theoretical guarantees.
>
> [1] On the convergence of encoder-only shallow transformers. NeurIPS.

---

### Decision · Program_Chairs · 2025-09-17

**Decision:**

Accept (poster)

**Comment:**

This paper proposes a framework to schedule Agile Earth Observation Satellites (AEOSs) constellations. The reviewing team was overwhelmingly positive about the submission, mentioning only relatively minor issues, which could be addressed for the camera-ready.
Please incorporate all the feedback received from the reviewers in the revised version.